# Cell-intrinsic mechanisms of temperature compensation in a grasshopper sensory receptor neuron

Frederic A Roemschied[1,2], Monika JB Eberhard[3], Jan-Hendrik Schleimer[1,2], Bernhard Ronacher[2,3], Susanne Schreiber[1,2]*

[1]Institute of Theoretical Biology, Department of Biology, Humboldt-Universität zu Berlin, Berlin, Germany; [2]Bernstein Center for Computational Neuroscience Berlin, Berlin, Germany; [3]Behavioral Physiology Group, Department of Biology, Humboldt-Universität zu Berlin, Berlin, Germany

**Abstract** Changes in temperature affect biochemical reaction rates and, consequently, neural processing. The nervous systems of poikilothermic animals must have evolved mechanisms enabling them to retain their functionality under varying temperatures. Auditory receptor neurons of grasshoppers respond to sound in a surprisingly temperature-compensated manner: firing rates depend moderately on temperature, with average $Q_{10}$ values around 1.5. Analysis of conductance-based neuron models reveals that temperature compensation of spike generation can be achieved solely relying on cell-intrinsic processes and despite a strong dependence of ion conductances on temperature. Remarkably, this type of temperature compensation need not come at an additional metabolic cost of spike generation. Firing rate-based information transfer is likely to increase with temperature and we derive predictions for an optimal temperature dependence of the tympanal transduction process fostering temperature compensation. The example of auditory receptor neurons demonstrates how neurons may exploit single-cell mechanisms to cope with multiple constraints in parallel.

*For correspondence:
s.schreiber@hu-berlin.de

**Competing interests:** The authors declare that no competing interests exist.

**Reviewing editor**: Ronald L Calabrese, Emory University, United States

## Introduction

Changes in temperature considerably modulate physico-chemical processes and, consequently, also affect neural processing (*Schmidt–Nielsen, 1997*; *Robertson and Money, 2012*). The dependence of neural activity on temperature poses a particular challenge for animals without central heat regulation, like insects, who are permanently exposed to temperature fluctuations. These animals must have evolved intrinsic mechanisms at the behavioral, systems, or cellular level that help to circumvent temperature-induced behavioral modulations. Such compensatory mechanisms, however, may also come into play for homeothermic animals under pathological conditions, like fever or hypothermia in mammals.

Nevertheless, our understanding of generic design principles that enhance robustness to temperature fluctuations remains limited. The goal of this study is to identify mechanisms and limitations of cellular temperature compensation at the level of firing rates. We start from a characterization of the temperature dependence of neural responses in an insect auditory system, which we find to be surprisingly robust to temperature changes. The absence of network inputs to these receptor neurons (*Vogel and Ronacher, 2007*; *Clemens et al., 2011*) suggests that a cellular mechanism underlies the observed temperature compensation and hence raises the more general question to what extent temperature compensation can be achieved at the level of individual cells. Based on generic conductance-based models of spike generation, we then show that the experimentally observed degree of temperature compensation can be explained by physiological properties intrinsic to single cells despite a substantial dependence of ion channels on temperature.

**eLife digest** Warm-blooded animals—including mammals and birds—expend large amounts of energy in keeping their body temperature constant regardless of how hot or cold their environment is. By contrast, the body temperature of cold-blooded animals—including amphibians, reptiles, and insects—follows that of their surroundings. Cold-blooded animals must therefore have evolved a means to cope with the effects of changes in temperature, but exactly how they do this is not clear.

Now, Roemschied et al. have obtained new insights into this process by studying the nerve cells in grasshoppers that allow them to hear sounds. The auditory system of the grasshopper comprises sensory receptor neurons that are located on the abdomen of the insect. Sound waves move the tympanal membrane, which causes ion channels within the cell membranes to open. This enables the neurons to produce an electrical signal known as a spike.

Recordings from grasshoppers revealed that changing the outside temperature by up to 10°C affected the rate at which the neurons produced spikes by only about half the amount expected. Given that these neurons do not receive inputs from any other cells, this ability to withstand changes in temperature must be intrinsic to the neurons themselves. Consistent with this, computational modeling showed that while the activity of individual ion channels did indeed vary with changes in temperature, these changes in ion channel activity had little overall effect on the rate at which a neuron produced spikes.

Whereas it has previously been assumed that compensation for changes in temperature occurs at the level of networks of neurons, the work of Roemschied et al. reveals that such compensation can occur in individual cells, and that it need not require a lot of energy to be expended.

Temperature dependence is usually quantified by the so-called $Q_{10}$ value, which characterizes the relative change of a variable when temperature rises by 10°C. Several invertebrate species were found to have firing-rate $Q_{10}$ values above 2 (i.e., to double their neurons' firing rate), which is in line with the fact that many underlying biochemical processes also exhibit $Q_{10}$ values of two or more (*French and Kuster, 1982*; *Pfau et al., 1989*; *Warzecha et al., 1999*; *Hille, 2001*; *Spavieri et al., 2010*). In contrast, we found that grasshopper auditory receptor neurons on average increased their firing rate by only ~40–50% (corresponding to a $Q_{10}$ value of 1.4–1.5). Receptor responses are shaped by a cascade of two major steps (*Gollisch and Herz, 2005*)—(1) auditory transduction, which translates the vibrations of the tympanal membrane into receptor currents and (2) spike generation. Temperature compensation of the response must be achieved by compensatory mechanisms in these individual components or their combined output.

Based on a computational analysis, we first investigate how the second component, that is cellular spike generation in terms of the translation from input current to firing rate, can be temperature compensated in generic model neurons and identify conductances whose temperature dependence favors robustness. As energy efficiency of signaling is an important constraint (*Attwell and Laughlin, 2001*; *Niven and Laughlin, 2008*), we also resolve whether the identified mechanisms for temperature compensation come at an additional metabolic cost and identify the key parameters of temperature dependence that increase energy efficiency of action-potential generation as well as of the maintenance of the resting potential. Moreover, we show that information transfer (via spike rates) is fostered by temperature increments.

Second, we combine spike generation with a phenomenological model of mechanotransduction and predict properties of the temperature dependence of this nonlinear transformation that would allow for an efficient compensation in firing rates to the degree observed in our experimental data. As our model-based approach generalizes beyond the grasshopper system, our findings can be expected to reflect principles that could be implemented in many invertebrate and vertebrate species.

## Results

### Temperature dependence of locust auditory receptor neuron responses to acoustic stimuli

Based on recordings of auditory receptor neurons in the metathoracic ganglion of the grasshopper *Locusta migratoria*, we quantified the dependence of the firing rate on temperature. *Figure 1A* shows

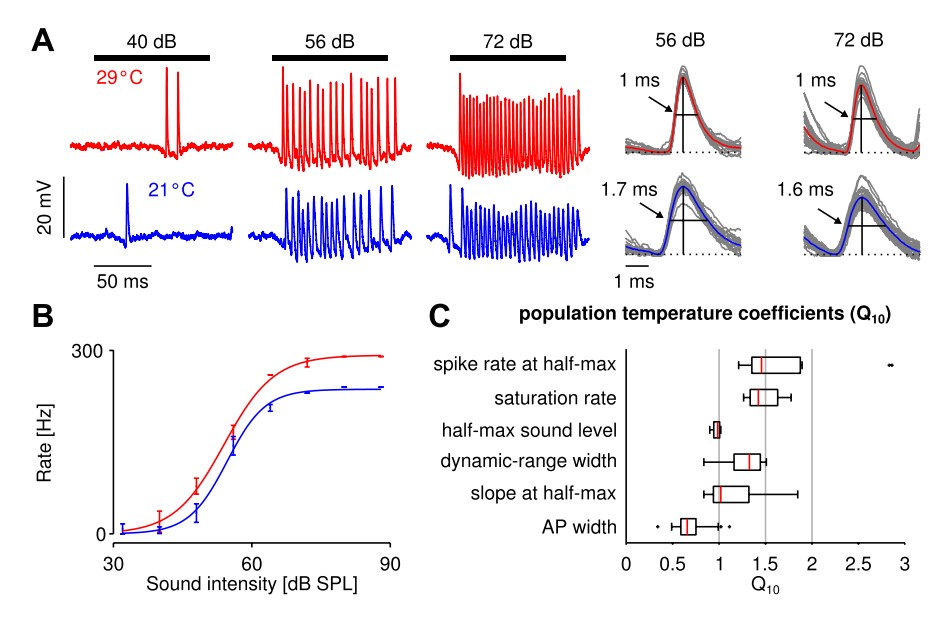

**Figure 1**. Cooling mildly affected electrophysiologically recorded firing rates generated by auditory receptor neurons in response to sound. (**A**) Voltage traces at 29 and 21°C for one neuron (red and blue lines, respectively). Black horizontal lines mark time intervals of stimulus presentation; stimulus intensity as indicated. (**B**) Firing-rate as a function of sound intensity was well described by sigmoidal functions at both temperatures (same neuron as in **A**). The three sigmoidal parameters (saturation rate, sound level at half-maximum, and dynamic-range width) were extracted from fits to the experimental data. (**C**) Statistics of the measured temperature dependence,

$$Q_{10}(x) = \left( \frac{x(T + \Delta T)}{x(T)} \right)^{10/\Delta T}$$

, were computed for several quantities $x$. For a population of nine receptor neurons all three parameters of the sigmoidal function, as well as the spike rate and slope at the cold half-maximum were temperature compensated (median $Q_{10} \in [1,1.5]$, see also *Figure 1—figure supplement 2* and *Figure 1—figure supplement 3*).

The following figure supplements are available for figure 1:

**Figure supplement 1**. Temperature calibration curve.

**Figure supplement 2**. Sound-intensity resolved p-values of statistical differences between firing rates at the two temperatures.

**Figure supplement 3**. Statistical analysis of Q10 values.

voltage responses to stimulation at three different sound intensities and two different temperatures, as well as spike shapes. Interestingly, spike rates at a given sound intensity did not differ much between the low and high temperatures and mildly increased from low to high temperature, while spike width decreased. In general, firing rates of grasshopper receptor neurons are relatively high, saturating only at several hundreds of Hz. At a given temperature, the transfer function, that is the firing rate as a function of sound intensity, has a sigmoidal shape (*Figure 1B*). Three parameters (saturation rate, half-max sound level, and dynamic-range width) are sufficient to capture the experimental transfer functions ($R^2 > 0.95$ for all response curves). Comparing transfer functions at the two different temperatures revealed that their temperature dependence was surprisingly low (*Figure 1C*): all corresponding median $Q_{10}$ values were below 1.5; the sound intensity at half-maximal response (half-max sound level) remained almost unchanged, as did the median of the slope at half-max sound level. The width of action potentials, in contrast, was lower at the higher temperature.

Compared to the temperature dependencies previously observed in other species like moth auditory receptor neurons, locust stretch receptors, and fly H1 neurons and photoreceptors ($Q_{10}$(spike rate) ~2, *Pfau et al., 1989*; *Coro and Perez, 1990*; *Warzecha et al., 1999*; *Tatler et al., 2000*), the grasshopper responses were temperature compensated; the dependence was similar to what has been referred to

as *warm-insensitive* in hypothalamic neurons of the rat ($Q_{10}$(spike rate) ~1.3, *Curras and Boulant, 1989*). To understand this low dependence of receptor neuron firing rate on temperature, we next turned to mathematical modeling.

## Influence of temperature on spike generation in single-neuron models

The grasshopper auditory periphery consists of a relatively simple feed-forward network, in which the receptor neurons constitute the first layer. Receptor neurons are known to respond to vibrations of the tympanal membrane, but they do not receive input from the neuronal network. Mechanisms of temperature compensation must hence be cell-intrinsic. To resolve which intrinsic processes can be sufficient to account for the observed degree of temperature compensation, we first focussed on spike generation, leaving mechanotransduction aside. We analyzed in generic model neurons, how the temperature dependence of ionic conductances mediating spike generation can reduce the dependence of firing rate on temperature.

We used the Connor–Stevens model (*Connor et al., 1977*; *Dayan and Abbott, 2005*) to simulate a type I spike generation process (*Izhikevich, 2007*) as it is assumed for grasshopper receptor neurons (*Benda, 2002*). Besides a sodium and a leak conductance ($g_{Na}$ and $g_L$), this model comprises a delayed-rectifier and an A-type potassium conductance ($g_K$ and $g_A$), which are both known to be present in the grasshopper nervous system (*Ramirez et al., 1999*). Temperature dependence was assumed to affect the opening and closing rates of all gates of the three ion channel types (i.e., the $m$ and $h$ gates for $g_{Na}$, $n$ gates for $g_K$, $a$ and $b$ gates for $g_A$), as well as their peak conductances ($\overline{g}_{Na}$, $\overline{g}_K$, $\overline{g}_A$) and that of the leak conductance, $\overline{g}_L$. For a systematic analysis (refers to (*Prinz et al., 2003*)), we independently varied the temperature dependence of these parameters (comprising a total of nine) within physiologically realistic ranges: $Q_{10}(x) \in [2.0, 4.0]$ for transition rates and $Q_{10}(\overline{g}_x) \in [1.2, 2.0]$ for peak conductances (*Partridge and Connor, 1978*; *Hille, 2001*). For each combination of parameters, the transfer function (input current to firing rate; i.e., the f-I curve) was computed at two temperatures: 18 and 28°C (*Figure 2A,B*).

To estimate the temperature dependence of a whole f-I curve, $Q_{10}$ values are not ideal, as they are defined as the ratio of firing rates at two different temperatures, which will be infinitely large for inputs that only elicit spikes at the higher, but not the lower temperature. To circumvent this bias, we assessed the temperature dependence of a model neuron as the root-mean-squared difference between the firing rates at the two temperatures (mean taken across input currents), normalized by the mean rate elicited at the colder temperature. We refer to this quantity as RMSD. It reflects the average relative change in firing rate with temperature. Note that a $Q_{10}$ value of 1.5 is hence comparable to an RMSD of 0.5 (50% relative change). Across all model combinations, the relative change, RMSD, was distributed between 0.22 and 2.14, with a median of 0.68 (*Figure 2D*). This means that, intuitively, the median of the average change in firing rate of an f-I curve was 68%. The analysis showed that the effect of temperature on spike generation depended strongly on the specific temperature dependence of the ionic conductances. A fraction of models (18%) exhibited temperature compensation with relative changes in firing rate comparable to those found experimentally (RMSD <0.5). This result shows that a low dependence of firing rate on temperature is feasible despite a substantial (and hence realistic) dependence of the individual conductances on temperature.

Next, we asked which of the nine parameters (i.e., the temperature dependence of peak conductances and transition rates) most affected the dependence of firing rate on temperature. To this end, we performed a systematic sensitivity analysis in the nine-dimensional space of all possible parameter combinations. We created—for each parameter—a distribution of local changes in RMSD induced by changes in that parameter. Specifically, this distribution captured changes in RMSD between all neighboring points along a specific parameter's dimension. Each distribution sampled the whole parameter space (i.e., all possible combinations of the other parameters), also see 'Materials and methods'. The *impact* of a given parameter on the temperature dependence of the f-I curve was then defined as the median of its specific distribution, directly relating the impact of a parameter to its quantitative effect on RMSD. *Figure 2C* depicts the impact values of all $Q_{10}$ parameters on the RMSD. The sum of all absolute values of impacts is normalized to unity. The impact sign, that is whether an increase in a parameter on average led to an increase or decrease in the observable, is indicated by + and −, respectively. The analysis revealed that the largest impact on the RMSD was exerted by three parameters of potassium channels: temperature dependence of the delayed-rectifier potassium kinetics, $Q_{10}(n)$, and the A-type and delayed-rectifier potassium peak conductances, $Q_{10}(\overline{g}_A)$ and $Q_{10}(\overline{g}_K)$, respectively. The impacts of both potassium channel peak conductances were negative (i.e., increases in their $Q_{10}$ values

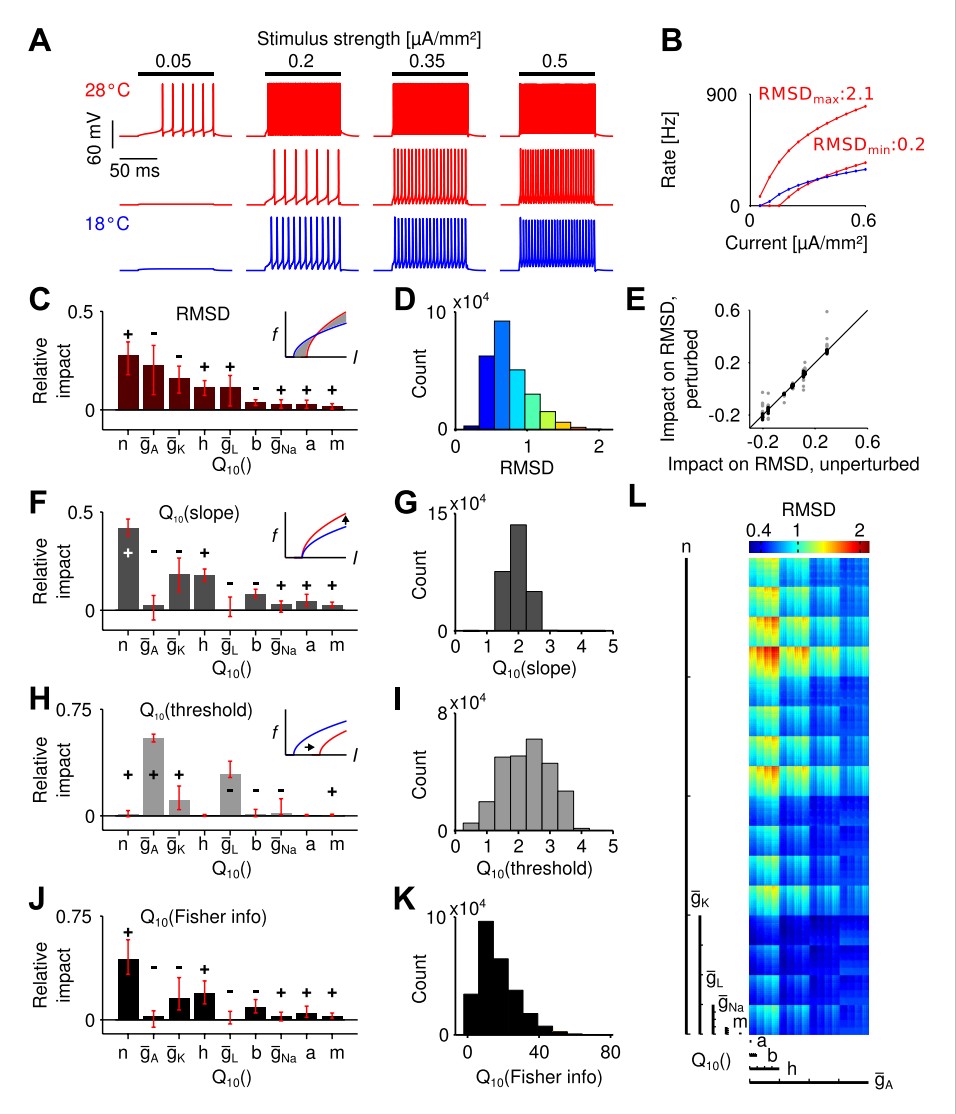

**Figure 2**. Temperature dependence of spike generation in a conductance-based neuron model. (**A**) Voltage responses to step current stimuli of different amplitudes; blue: 18°C (the reference temperature), red: 28°C. Top trace corresponds to a model with strongly temperature-dependent firing rate, middle trace to a temperature-compensated model. (**B**) f-I curves at both temperatures, corresponding to the examples shown in **A**. (**C**) Results of the sensitivity analysis for the RMSD. The largest impact is exerted by temperature dependencies of the potassium conductances ($Q_{10}(n)$, $Q_{10}(\bar{g}_A)$, and $Q_{10}(\bar{g}_K)$). Signs +/− indicate the qualitative impact (see main text for details). (**D**) Distribution of the RMSD, across all models. Note that a $Q_{10}$ of 1.5 corresponds to an RMSD of ~0.5 (50% relative change). (**E**) Parameter impacts on RMSD were robust against ±20% perturbation of the model's peak conductances at 18°C (black symbols: perturbations of individual peak conductances; grey symbols: all combinations of ±20% changes to the four peak conductances). (**F**) Results of the sensitivity analysis for the temperature dependence of the slope of f-I curves. (**G**) Distribution of $Q_{10}$ values of the slope across all models. (**H** and **I**) as panels (**F**) and (**G**), but for the threshold of the f-I curves. (**J** and **K**) Sensitivity analysis of information transfer. For two basic noise models (Poissonian and input-independent Gaussian), the temperature dependence of firing rate-based information transfer $J$ is related to that of the slope of the f-I curve: $Q_{10}(\langle J \rangle) = [Q_{10}(\text{slope})]^4$. The conductance parameters with highest impact were very similar to those of the changes in slope (compare to panel **F**). Information transfer increased with temperature for all models. (**L**) Visualization of the RMSD for the parameter space spanned by the temperature dependencies of the Connor-Stevens model based on dimensional stacking. Axes order was chosen according to the impact ranking as presented in (**C**); color code as in (**D**).

*Figure 2. Continued on next page*

*Figure 2. Continued*

The following figure supplements are available for figure 2:

**Figure supplement 1**. Temperature compensation in the Traub-Miles model.

---

decreased the RMSD), while the potassium activation $Q_{10}(n)$ had a positive impact (i.e., increases in its temperature dependence increased the RMSD).

To confirm that the results do not strongly depend on the specific choice of peak conductances in the Connor–Stevens model, we tested 24 alternative models with changes of ± 20% in the peak conductances of sodium, both potassium, and leak channels. The impact ranking across those models was highly similar to the ranking in the original Connor–Stevens model (*Figure 2E*) and we conclude that our results are robust. Moreover, we note that our results are not unique to the Connor–Stevens model. An analysis of a structurally different Traub–Miles model (*Traub et al., 1991*; *Benda, 2002*) showed that an equally low temperature dependence is possible (*Figure 2—figure supplement 1*).

On a side note, a visualization of the RMSD across the complete nine-dimensional parameter space based on dimensional stacking is shown in *Figure 2L*, see 'Materials and methods' for details (*LeBlanc et al., 1990*; *Taylor et al., 2006*). Dimensional stacking maps the nine-dimensional space onto a two-dimensional representation with nine axes. Ordering of the axes is arbitrary and hence requires optimization to maximize visual information (*Taylor et al., 2006*). Here, we introduce a new way to determine optimal axes order, defined directly by the ranking of impact scores. Parameters with highest impact on the RMSD are depicted on large-scale axes and parameters with low impact on small-scale axes. The success of the ordering is reflected in the visually structured areas of equal RMSD. As only a subset of all parameters had relevant influence on the RMSD, optimal axes ordering led to a clear visual structure. In contrast, for arbitrary axes ordering visual structure would be hard to recognize.

## Temperature effects on f-I threshold and slope

As we saw, the temperature dependence of potassium channels plays a crucial role for temperature compensation. For a more detailed and intuitive understanding of the underlying mechanism, we next analyzed the transformation of the shape of f-I curves with temperature. Type I f-I curves, as they are found in the grasshopper, can be described by a square root function (*Izhikevich, 2007*). Temperature affects an f-I curve by shifting the curve horizontally (i.e., changing its threshold) and by changing its slope (which can also be termed gain). We used fits of the f-I curves by a square root model $f(I) = A \cdot \sqrt{I - I_0}$, based on the parameters $A$ (slope) and $I_0$ (threshold). With heating, the slope always increased (*Figure 2G*), while we found changes in both directions for the threshold (*Figure 2I*).

Temperature dependencies of the A-type potassium and leak peak conductances had the strongest influence on the threshold (*Figure 2H*). In contrast, the slope was most sensitive to the temperature dependence of the delayed-rectifier potassium channel, $Q_{10}(n)$ and $Q_{10}(\bar{g}_K)$, and the sodium channel inactivation, $Q_{10}(h)$ (*Figure 2F*). Beyond clarifying the specific effect of the aforementioned parameters on changes to the f-I curve, the analysis shows that temperature compensation (i.e., lower RMSD values) was usually achieved by modest increases in threshold balancing the effects of an increase in slope (*Figure 2B*).

Changes in slope also have direct implications for the ability to infer information about the sound intensity from the firing-rate output of receptor neurons. We hence quantified how the capacity to transmit information from input $I$ to firing rate $f$ changes with temperature. To this end we use Fisher information. Considering the average information transferred for a given interval of firing rates $[f_{min}, f_{max}]$, information transfer scales with the slope of the f-I curve and its temperature dependence hence with $Q_{10}(A)^4$ ('Materials and methods'). Consequently, the same parameters that had the largest impact on the slope–potassium channel rate ($Q_{10}(n)$) and peak conductances ($Q_{10}(\bar{g}_K)$, $Q_{10}(\bar{g}_A)$) and sodium channel inactivation ($Q_{10}(h)$)—also influenced information transfer most (*Figure 2J*). Overall, heating was advantageous for information transfer (*Figure 2K*).

## Influence of temperature on the energy efficiency of spike generation

Metabolic cost is increasingly recognized as an important constraint for neural function (*Attwell and Laughlin, 2001*; *Niven and Laughlin, 2008*) and is likely to have shaped the design of neural

systems—the more so if firing rates are large. In the grasshopper auditory periphery firing rates often exceed several hundreds of Hz, suggesting that metabolic cost may have played a role in the design of these cells. It is hence interesting to explore whether robustness to temperature changes compromises energy efficiency. To this end, we computed the energetic cost of spike generation and maintenance of the resting potential (*Figure 3A*). Cost was quantified in terms of the total sodium current (per action potential or per time, respectively). To assess the changes of energy consumption with temperature, energy use was characterized by its $Q_{10}$ value (i.e., the ratio of energetic cost at 28° and 18°) and averaged across input currents.

For the majority of models, the energetic cost of an action potential decreased with heating (93% of models, *Figure 3B*). On average, temperature-compensated spike generation models (25% of models with lowest RMSD) were slightly more costly than the most temperature-dependent models (25% of models with highest RMSD), see *Figure 3B*. Nevertheless, the minimum energy consumption was comparable in both groups. Resting cost was substantially lower than spiking cost, this trend increasing with larger firing rates. Sodium-current based resting cost tended to decrease with heating (77% of models). It was slightly lower for temperature-compensated models compared to strongly temperature-dependent models (*Figure 3C*).

The sensitivity analysis (performed in analogy to the analysis of temperature dependence of firing rate) revealed that the temperature-dependence of the sodium conductance ($Q_{10}(h)$ and $Q_{10}(\bar{g}_{Na})$) influenced energy consumption of spike generation the most (*Figure 3D*).

The impacts of conductance parameters on f-I curve temperature dependence and energy consumption were not significantly correlated in this case ($\rho = -0.23$, $p=0.56$). In particular, the key

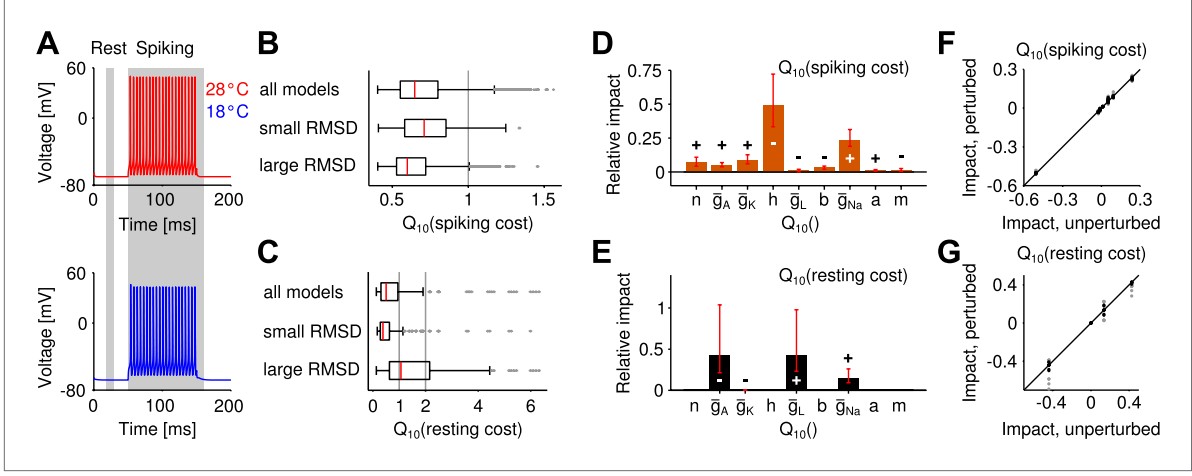

**Figure 3**. Temperature dependence of the metabolic cost (spiking and maintenance of the resting potential). (**A**) Illustration of the periods during which spiking cost and resting-potential cost were estimated in terms of the Na⁺ current (two corresponding examples at the higher and lower temperature; red and blue curves, respectively). Energy consumption during spiking was averaged per spike and across the suprathreshold parts of the f-I curve. (**B**) Distribution of the temperature dependence of the spiking cost ($Q_{10}$(spiking cost)). Top: all models; middle and bottom: distribution across the 25% of models with lowest and highest temperature dependence of firing rate (RMSD), respectively. Spiking cost decreased at higher temperature for the majority of models. In particular, values of minimal energy consumption at the higher temperature were similar for the subgroups of models with lowest and highest temperature dependence. (**C**) Distribution of the temperature dependence of the resting cost ($Q_{10}$(resting cost)), analog to (**B**). Resting-potential cost decreased for the majority of models; top, middle and bottom panels comprising subgroups of models as in (**B**). (**D**) Temperature dependence of firing rate and spiking energy consumption are determined by different sets of conductance parameters. While potassium-channel temperature dependencies have the largest impact on firing rate (*Figure 2C*), the energy consumption per spike was predominantly determined by sodium-channel temperature dependence. Faster sodium inactivation ($Q_{10}(h)$) and lower peak sodium conductance ($Q_{10}(\bar{g}_{Na})$) fostered energy efficiency at higher temperature. (**E**) Parameters that reduce resting energy at high temperature also reduce RMSD (same sign of the impact values as in *Figure 2C*). (**F** and **G**) Impacts on energy consumption were robust against ±20% perturbations of the model's peak conductances at 18°C (*Figure 2E*, 'Materials and methods').

The following figure supplements are available for figure 3:

**Figure supplement 1**. Alternative measures of metabolic cost and energy efficiency of spiking.

**Figure supplement 2**. Potassium-current based resting cost.

parameters of largest influence on these features belonged to different channel groups: potassium channels in case of temperature compensation and sodium channels in case of energy efficiency of spike generation. We verified that the large influence of sodium channels was not biased by our sodium-current-based definition of metabolic cost. Three alternative measures—two quantifying energy efficiency based on the separability of sodium and potassium currents (*Alle et al., 2009*), the other defined by the total potassium current—all confirmed the temperature dependence of sodium channel inactivation $Q_{10}(h)$ as the most influential parameter for spiking cost (*Figure 3—figure supplement 1*).

The sodium-current based resting cost was qualitatively influenced in a similar way as temperature dependence of firing rate (*Figure 3E*): for all four relevant $Q_{10}$ parameters a reduction of resting cost co-occurred with a reduction in temperature dependence of firing rate (same sign of corresponding impacts, *Figure 2C*). For a potassium-current based resting cost the temperature dependence of leak channels had the dominant impact (*Figure 3—figure supplement 2*). In contrast to the sodium-current based resting cost, the potassium-current-based cost was larger at higher temperatures. For the majority of models, inactivation of A-type potassium channels was lower at the higher temperature (due to a more negative resting potential) and hence increased the A-type current.

In summary, striving for temperature compensation does not have to compromise a neuron's energy efficiency. Both results (for spiking and resting cost) generalize beyond the specific choice of peak conductances in the Connor–Stevens model (*Figure 3F,G*).

## Model-based inference of the auditory transduction function

In the previous paragraphs, we have shown that spike generation by itself can achieve a remarkably high invariance to temperature changes. The receptor neurons, however, have an additional processing stage involved in transferring the external input signal to a firing-rate response: the so-called transduction mediated by the mechanosensitive channels in the vicinity of the tympanal membrane. Transduction precedes spike generation and translates vibrations of the tympanal membrane caused by the sound pressure wave into receptor currents through these channels, which in turn drive spike generation (*Gollisch and Herz, 2005*). This mechanism may also contribute to temperature compensation. We therefore explored how temperature compensation can benefit from linking spike generation with the transduction process.

Little is known about the temperature dependence of transduction in the grasshopper. Here, we combined the computational analysis of spike generation with the experimental findings for the sound-intensity to firing-rate relation in order to predict on a phenomenological level which features of the auditory transduction and its temperature dependence would improve temperature compensation. The experimentally measured receptor neuron responses to sound stimuli $r = \rho(I_{dB})$ can be expressed as a cascade of mechanosensory transduction ($I_C = \theta(I_{dB})$, with current $I_C$ and sound intensity $I_{dB}$) and spike generation $\varphi(I_C) : r = \varphi(I_C) = \varphi(\theta(I_{dB}))$.

Let us think of the Connor–Stevens spike generation model at the colder temperature (illustrated in *Figure 4A*, blue curve). When combined with an (upstream) nonlinear translation of sound intensity to current (*Figure 4B*, blue curve), it yields the full sound-intensity to firing-rate relation (*Figure 4C*, blue curve), which corresponds to the quantity measured experimentally. Hence, if the receptor neuron response as a function of sound intensity is known from experimental data and we assume a specific spike generation model (i.e., a specific f-I curve), we can mathematically infer the transduction nonlinearity that gives the best match to the experimentally recorded sound-intensity to firing-rate relation by nonlinear regression (*Figure 4C*, blue curves). We use the term reverse-engineering for this approach. It can be used at the higher temperature as well and enables us to derive for each of the more than 260.000 spike generation models *the corresponding ideal* transduction curve that best matches the experimentally recorded sound-intensity to firing-rate relation at this higher temperature (*Figure 4B,C*, red curves). Comparing the reverse-engineered transduction curve at the colder temperature to the reverse-engineered curve at the higher temperature (for each spike generation model), allows us to identify trends in the temperature dependence of mechanotransduction, which would foster a temperature robustness of the firing rate.

Specifically, we exploited the fact that experimental response curves (firing rate to sound intensity) were well fitted by a sigmoidal function (*Figure 1C*) and also assumed a sigmoidal shape for the transduction curves (*Hudspeth et al., 2000*; *Fisch et al., 2012*). Accordingly, we reverse-engineered the ideal transduction sigmoid at the warmer temperature for each spike generation model (e.g., those contributing to *Figure 2C*), so that the resulting sound-intensity to firing-rate relation best matched a

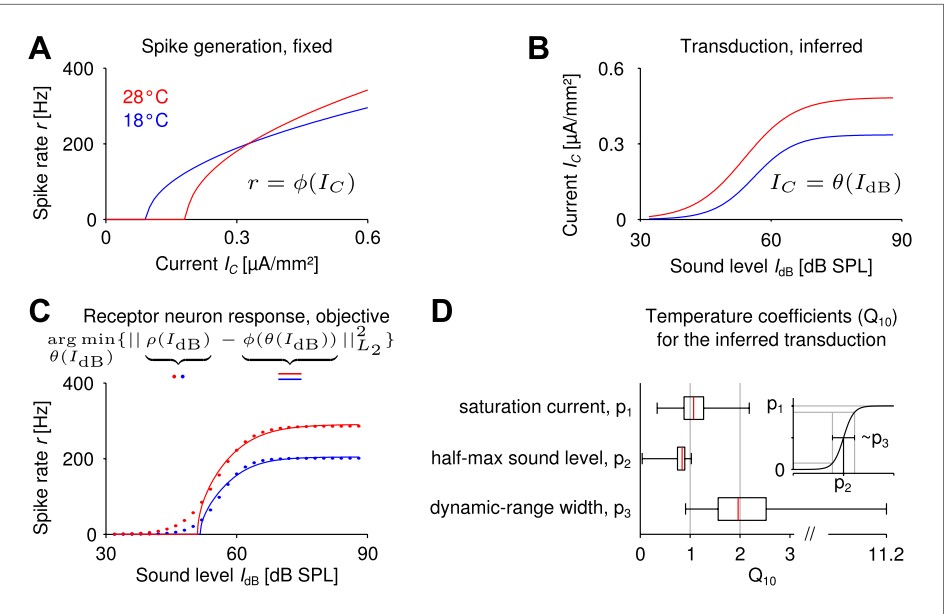

**Figure 4**. Reverse engineering mechanosensory transduction functions that favor temperature compensation of firing rate. (**A**) Example of a model-based f-I curve, denoted $r = \varphi(I_C)$, at two temperatures. (**B**) Example of a sigmoidal transduction function converting sound intensity to current, $I_C = \theta(I_{dB})$. (**C**) Representative receptor neuron responses, $r = \rho(I_{dB})$, at two temperatures (dotted lines), as well as a receptor neuron response, $r = \varphi(\theta(I_{dB}))$, 'constructed' from the f-I curve in **A** and the transduction function in **B**. For each model of spike generation, the optimal transduction function, $I_C = \theta(I_{dB})$, minimizing the error between the corresponding 'constructed' receptor neuron response and the representative receptor neuron response (dotted line in **C**) was derived. (**D**) The statistics of temperature dependence of the optimal transduction functions, $I_C$, across all models. The width of the dynamic range depended most on temperature and increased with heating for nearly all optimized transduction functions. But the temperature dependence of the saturation current and the half-maximum sound intensity (mainly when decreasing with temperature) were also found to contribute to temperature compensation. Note that the ranges marked by the whiskers cover all data (including outliers) in this plot.

representative receptor neuron response (*Figure 4C*, for details on representative receptor neuron response, 'Materials and methods'). The model response curves $r = \varphi(\theta(I_{dB}))$ matched the experimental representative response curve very well ($R^2 > 0.98$ for 99.5% of all models). Temperature dependence of the reverse-engineered transduction was then quantified based on the $Q_{10}$ values (i.e., the relative changes with temperature) of the three parameters that define each transduction sigmoid: saturation current, half-maximum sound intensity, and dynamic-range width (*Figure 4D*).

Evaluating the distribution of changes of the ideal transduction curves with temperature across all spike generation models, we found that the largest temperature dependence of these 'matching' transduction curves was to be expected for their dynamic-range width (median $Q_{10} \sim 2$). In addition, changes in saturation current and half-maximum sound level also contributed to fostering temperature compensation of firing rate (*Figure 4D*). These results show in particular that a suitable temperature dependence of the transduction process can support temperature compensation, even in cases where spike generation is less temperature robust.

## Discussion

We studied the temperature sensitivity of the firing rate in individual neurons of the grasshopper auditory periphery and found responses to auditory stimulation to be surprisingly temperature compensated. Based on biophysically-motivated neuron models, we identified mechanisms that account for the experimentally observed *cell-intrinsic* temperature compensation. Our theoretical analysis suggests that spike generation itself can be relatively temperature insensitive, even though the conductances involved are affected by temperature changes. Importantly, mechanisms increasing the robustness to temperature changes need not compromise the energy efficiency of action-potential

generation nor the resting cost. In general, the capacity to transmit rate-based information of sound intensity moderately increased with temperature due to the increase in steepness of the f-I curve. We also predict optimal temperature dependencies of the tympanum-mediated transduction process from sound to receptor current that contribute to temperature compensation. On a side note, we introduced a computationally efficient way to optimize visualization of model-derived observables in a high-dimensional parameter space in the context of dimensional stacking (*LeBlanc et al., 1990*; *Taylor et al., 2006*).

## Temperature dependence of firing rate in single neurons

Neuronal processing is significantly challenged by variation in temperature due to the changes in chemical and physical processes. In many neurons across invertebrates and vertebrates, firing rate has been observed to at least double with increases of temperature, corresponding to $Q_{10}$ values of two or above (*French and Kuster, 1982*; *Coro and Perez, 1990*; *Warzecha et al., 1999*). It was hence surprising to observe that neurons in the auditory periphery of grasshoppers show $Q_{10}$ values on the order of 1.5 and consequently are remarkably temperature compensated (*Hazel and Prosser, 1974*; *Boulant and Dean, 1986*). The temperature robustness of these neurons is hence comparable to that of 'warm-insensitive' neurons in the mammalian brain (*Curras and Boulant, 1989*).

Temperature compensation has been studied in the context of neurons embedded in a network in a variety of systems (see, e.g., *Wechselberger et al., 2006*; *Tang et al., 2010*; *Robertson and Money, 2012*). Temperature compensation in grasshopper receptor neurons, by comparison, must be based on cell-intrinsic processes. A similar temperature compensation that must be based on a single-cell mechanism has so far—to our knowledge—only been described experimentally for tarsal hairs in the locust (*Miles, 1985*).

The cell-intrinsic mechanisms identified in our computational study attribute the observed robustness to a balancing of opposing processes. Phenomenologically, a rise in slope (consistent across the whole parameter range explored) is compensated for by an increase in the threshold of the f-I curve, minimizing the effect of temperature across a broader range of inputs. While alterations in threshold can be produced by changes in peak conductances of ion channels, they have also been described experimentally by heating in invertebrate systems (*Burkhardt, 1959*; *Abrams and Pearson, 1982*; *Kispersky et al., 2012*) in agreement with our observations. Biophysically, a heating-induced increase in the speed of repolarizing gating kinetics is opposed by an increase in peak potassium conductances which promote a more negative resting potential (as can be derived from *Equation 2*). Although the balancing is not perfect, average deviations on the order of not more than 50% can be easily achieved with strongly temperature-dependent conductances (in particular, $Q_{10} \in [2; 4]$ for all activation- and inactivation rates) for ~18% of the models. The temperature dependence of both delayed-rectifier and A-type potassium channels has a particularly large impact on temperature compensation. This matches experimental observations in neurons of the pancreas of mice (*Xu et al., 2006*) and molluscan neurons as well as previous simulations of an extended Hodgkin–Huxley model (*Rush and Rinzel, 1995*) and is consistent with the effect of peak conductances on firing rate, for example (*Schreiber et al., 2004*). Our results also hold for reference models quantitatively different from the original Connor-Stevens model—both for the total fraction of temperature-compensated models RMSD <0.5) and the strong influence of potassium channel dynamics on the temperature dependence of firing rate (*Figure 2E*), 'Materials and methods' for details. These findings show that our results generalize beyond the specific quantitative choice of peak conductance parameters of the Connor–Stevens model. This is further supported by the fact that a structurally different Traub–Miles model could also exhibit a temperature dependence of firing rate as low as that described for the Connor–Stevens model.

## Energy efficiency and information transfer

For auditory receptor neurons in the grasshopper energy efficiency of spike generation is likely to be a relevant factor, also see *Niven and Farris (2012)*. Firing rates in these cells approach 400 Hz and likely entail a high total cost of electrical signaling. Our results, however, show that temperature compensation need not impair energy efficiency of spike generation nor of maintenance of the resting potential. The rate of sodium channel inactivation (*Figure 3D*) proved to be most relevant in setting the energy consumption per spike generated, which is consistent with simulations and dynamic clamp experiments in various model systems (*Alle et al., 2009*; *Hasenstaub et al., 2010*; *Sengupta et al., 2010*). We demonstrated that energy efficiency improved with heating for a wide range of temperature

dependencies of ion channels, as was previously described for a model with fixed $Q_{10}$ values (*Yu et al., 2012*). A fast sodium inactivation limits the duration of the spike; this was consistent with the experimental data, as spike width decreased with heating. Most importantly, the key parameters regulating the energy efficiency of spiking were different from those regulating temperature compensation of firing rate (*Figure 2C*, *Figure 3D*). The results could be confirmed for alternative measures of energy (separability of sodium and potassium currents as well as the total potassium current). Apart from confirming the role of sodium channel inactivation, these analyses substantiated that the delayed-rectifier potassium channel kinetics (which were most influential to the robustness of firing rate) did not substantially contribute to metabolic costs based on potassium currents. Overall, our analysis focusses on a major source of metabolic cost: the flow of $Na^+$ and $K^+$ ions which on larger time scales can be compensated by the Na-K-ATPase. For completeness it should be noted, however, that in living cells metabolic costs can also arise from the flow of other ions not included in our study, like $Ca^{2+}$.

Summarizing the considerations on metabolic cost, we find most noteworthy that from an evolutionary perspective, the relevant features—robustness of firing rate to temperature changes and reduction of metabolic cost—could both be achieved in parallel.

Nevertheless, temperature compensation and energy efficiency would be of little use if the fundamental function of information transmission was impaired. Our sensitivity analysis revealed that higher temperatures are also advantageous for the transmission of information about sound intensity. This conclusion is based on our finding that the slope of the f-I curve increased with heating. The capacity to transmit information was affected most by the temperature dependence of the delayed-rectifier potassium conductance. These data, however, need to be interpreted with care. We cannot exclude that channel-type specific stochastic dynamics further influence information transfer in ways not captured by our approach. Implementing the specific stochastic dynamics for the whole parameter space of more than 260.000 models, however, goes beyond the scope of this study and merits future investigation.

## Optimization of the input layer is important

Auditory receptor neurons in the grasshopper constitute the bottom layer of a feedforward network: approximately 80 receptor neurons converge to ~15 local neurons, which in turn project to ~20 ascending neurons (*Vogel and Ronacher, 2007*). All auditory input passes through this peripheral network, which preprocesses information and extracts behaviorally-relevant features (*Clemens et al., 2011*). The large investment into high firing rates and a comparatively high redundancy between neurons in this layer (*Machens et al., 2001*) also increases the need for energy-efficient spike generation. Optimization of receptor neurons in terms of temperature compensation hence seems a reasonable 'strategy', as all effects of temperature on receptor neurons will be passed on to downstream neurons, where they may multiply. Although we currently do not know to which extent other parts of the auditory system are compensated, it is likely to 'pay off' to constrain the effects of temperature in the initial stages. Downstream neurons, in contrast, may be expected to adopt different strategies, as they can make use of different mechanisms: balancing of inhibition and excitation for robustness to temperature changes (*Robertson and Money, 2012*) as well as an increase in population and temporal sparseness for energy efficiency and information transfer (*Clemens et al., 2012*).

## Temperature dependence of the transduction process

While the considerations above refer to spike generation, little is known about the temperature dependence of the preceding transduction process. One hypothesis is that a change in the half-maximum sound intensity of the transduction process could foster temperature compensation in firing rate. Our computational analysis shows that, indeed, a slight shift of transduction to lower sound intensities with higher temperatures may be favorable. Such a shift would occur if the amplitude of the tympanal vibration increased with heating and a stimulus of given intensity hence opened more transduction channels. However, the computationally-derived changes are relatively moderate (*Figure 4D*). This is consistent with the observation that the tympanal vibration in cicadas is relatively temperature independent (*Fonseca and Correia, 2007*).

The other two parameters characterizing transduction (saturation current and dynamic-range width) reflect properties of the transduction channels (i.e., their peak conductance and activation range, respectively). The increase in dynamic-range width with heating can be interpreted as a decrease in gating force in a gating-spring model for the transduction as proposed for mechanosensory

transduction in bullfrog saccular hair cells (*Howard and Hudspeth, 1988*). Again, increases in temperature are advantageous, because the gating-force magnitude is inversely related to transduction accuracy (*van Netten et al., 2003*). Depending on the spike-generation process, increases or decreases in the saturation current foster temperature compensation. The former may directly arise from the temperature dependence of the transduction channels' maximal conductance. The latter may require additional heat-sensitive channels with a modulatory influence on the transduction process, such as thermosensitive transient receptor channels (TRPA) (*Kang et al., 2012*), which in principle could down-regulate the saturation level of the transduction function via their increased calcium response (*Chadha and Cook, 2012*).

Generally, the analysis shows that the transduction process can contribute to temperature compensation. While spike generation alone is sufficient to mediate robustness of the firing rate, a matched temperature dependence of the transduction process may allow for more flexibility in the 'choice' of spike generation parameters, including the possibility to meet additional constraints.

## Conclusions

Altogether, our data show that auditory receptor neurons in the grasshopper represent an example of remarkable cell-intrinsic temperature compensation in the absence of network effects. Our computational analysis clarifies that spike generation alone can achieve this high degree of invariance of firing rate to temperature changes. The identified mechanisms generalize to spike generation in other cell types. Moreover, additional nonlinear processing by static nonlinearities (here interpreted as the transduction process involving the tympanal membrane, but on a wider scope also reflecting properties of synaptic transmission) may foster temperature compensation, if well matched with the temperature dependence of spike generation. Overall, the dependence of neuronal processing on temperature merits further investigation, in particular as temperature fluctuations are an oftentimes underestimated variable in mammalian systems too.

## Materials and methods

### Experimental animals and electrophysiology

Experiments were performed on adult *L. migratoria*, obtained from a commercial supplier and held at room temperature (22–25°C). Intracellular recordings from auditory neurons within the metathoracic ganglion were conventionally conducted as described elsewhere (*Franz and Ronacher, 2002*; *Wohlgemuth and Ronacher, 2007*), using glass microelectrodes filled with a 3–5% solution of Lucifer yellow in 0.5 M LiCl. Neuronal responses were amplified (BRAMP-01; npi electronic GmbH, Tamm, Germany) and recorded by a data-acquisition board (BNC-2090A; National Instruments, Austin, TX) with 20 kHz sampling rate. To control for temperature, the preparation was placed directly on a Peltier element connected to a 2 V battery and a potentiometer. Temperature was monitored and recorded with a digital thermometer (GMH 3210, Greisinger electronic GmbH, Regenstauf, Germany) connected to a NiCr-Ni-thermoelement (GTF 300, Type K, Greisinger electronic GmbH, Regenstauf, Germany). For each experiment, recordings were conducted first at a fixed higher tissue temperature (in the range of 28–29°C), then the preparation was cooled down to a lower temperature (in the range of 21–23°) and recordings were repeated.

To control for differences between the temperatures of the Peltier element and the tissue at the inner side of the tympanal membrane at the attachment site of receptor neurons, the dependence between those variables was measured directly and used for calibration (*Figure 1—figure supplement 1*). The calibration shows that at the higher Peltier temperature (30°C) tissue temperature only reached 28°C (in the steady state) due to heat dissipation. After the cooling process the difference between Peltier and tissue temperature in the steady-state was less than 0.5°C. Moreover, cooling down proved to be slower in the tissue than at the Peltier element. In order not to underestimate $Q_{10}$ values, we took a conservative approach: Electrophysiological recordings started 3–5 min after induction of the temperature change. Tissue temperature was derived from the calibration curve at the onset of a recording (lasting 40 s). Although temperature may still have been subject to small changes during the recording, this procedure ensured that temperature changes (i.e., the difference between high and low temperature) were—at most—slightly underestimated, favoring larger $Q_{10}$ values. Consequently, the estimated $Q_{10}$ values constitute an upper bound. In contrast, we cannot exclude that real $Q_{10}$ values may even be slightly smaller, that is even more temperature compensated.

After completion of the recordings, Lucifer yellow was injected into the recorded cell by applying a hyperpolarizing current. Subsequently, the thoracic ganglia were removed, fixed in 4% paraformalde-hyde, dehydrated, and cleared in methylsalicylate. The stained cells were identified under a fluorescent microscope according to their characteristic morphology. Altogether, nine receptor neurons were recorded in eight preparations.

## Acoustic stimulation

To obtain spike rate vs intensity curves (response curves), we used acoustic broad band stimuli (100 ms duration, 1–40 kHz bandwidth) repeated five times each at 8 intensities, rising from 32 to 88 dB SPL. Acoustic stimuli were stored digitally and delivered by a custom-made program (LabView 7 Express, National Instruments, Austin, TX). Following a 100 kHz D/A conversion (BNC-2090A; National Instruments, Austin, TX), the stimulus was routed through a computer-controlled attenuator (ATN-01M; npi electronic GmbH, Tamm, Germany) and an audio amplifier (Pioneer stereo amplifier A-207R, Pioneer Electronics Inc., USA). Acoustic stimuli were broadcast unilaterally by speakers (D2905/970000; Scan-Speak, Videbæk, Denmark) located at ± 90° and 30 cm from the preparation. Sound intensity was calibrated with a half inch microphone (type 4133; Brüel & Kjær, Nærum, Denmark) and a measuring amplifier (type 2209; Brüel & Kjær, Nærum, Denmark), positioned at the site of the preparation.

## Analysis of experimental data

Experimental spike times were extracted from the digitized recordings by applying a voltage threshold above background noise level. Mean spike rates were calculated for each intensity to obtain response curves (spike rate $r$ vs sound intensity $I_{dB}$) per neuron, stimulation side, and temperature. We fit a three-parameter sigmoid to each response curve, $r = \rho\left(I_{dB}\right) = r_{sat}\left/\left(1 + \exp\left(-\frac{I_{dB} - I_{50,\rho}}{w_\rho}\right)\right)\right.$, with saturation spike rate $r_{sat}$, half-maximum sound intensity $I_{50,\rho}$, and dynamic-range width $w_\rho$.

## Quantification of temperature effects

Unless noted otherwise, temperature dependence of a given observable $x$ was quantified by the temperature coefficient

$$Q_{10}\left(x\right) = \left(\frac{x\left(T + \Delta T\right)}{x\left(T\right)}\right)^{10/\Delta T} .$$

$Q_{10}(x)$ is the factor by which $x$ changes after a temperature increase of 10°C. $Q_{10} > 1$ and $Q_{10} < 1$ indicate an increase or decrease, respectively, of $x$ with heating, while $Q_{10} = 1$ indicates perfect temperature invariance. For plots of $Q_{10}$ values data points were presented as outliers when they fell outside the interval $[q_1 1.5 \cdot iqr, q_3 + 1.5 \cdot iqr]$, with the 25th and the 75th percentile defining $q_1$ and $q_3$ and an interquartile range $iqr = q_3 − q_1$.

## Temperature dependence of action-potential width

We also quantified the temperature-dependence of action-potential width at half-maximum amplitude, $Q_{10}$(AP width), for every neuron during the stimulus period, separately at each stimulus. *Figure 1C* shows the distribution of $Q_{10}$(AP width) pooled across all stimulus amplitudes (median 0.66). Our results qualitatively agree with the finding of broader action potentials at lower temperatures reported for various vertebrate and invertebrate neurons (*Thompson et al., 1985*; *Bestmann and Dippold, 1989*; *Janssen, 1992*; *Gabbiani et al., 1999*). Further, our results agree quantitatively with those reported for locust motor neurons and locust L-neurons (*Burrows, 1989*; *Simmons, 1990*).

## Adaptation

We checked that our results on the temperature dependence of firing rate were not compromised by the effects of adaptation. To this end, we re-analyzed the experimental data, separately focusing on the early phase of stimulus presentation (10–40 ms post stimulus onset), and the late phase (70–100 ms post stimulus onset). Effects of adaptation were reflected in a ratio of the respective parameter values (early-versus-late phase) that differed from one. While individual characteristics of experimentally measured firing-rate curves were subject to adaptation (e.g., the slope at half-maximum sound level was steeper early on and shallower in the later part), the early-to-late ratios did not significantly change with temperature.

## Single-neuron models

### Model definition

We performed simulations of neuronal membrane potential dynamics using a single-compartment Connor–Stevens model (original model described in *Connor et al., 1977*; parameters taken from *Dayan and Abbott 2005*):

$$C_m \frac{dV}{dt} = I_C - I_L - I_{Na} - I_K - I_A$$

$$= I_C - \bar{g}_L \cdot (V - E_L) - \bar{g}_{Na} \cdot m^3 h \cdot (V - E_{Na})$$

$$- \bar{g}_K \cdot n^4 \cdot (V - E_K) - \bar{g}_A \cdot a^3 b \cdot (V - E_A).$$

The original model was defined at a temperature of 18°C, where the parameters take the following values: peak conductances $\bar{g}_L = 0.003$, $\bar{g}_{Na} = 1.2$, $\bar{g}_K = 0.2$, $\bar{g}_A = 0.477$ (mS/mm²), and reversal potentials $E_L = -17$, $E_{Na} = 55$, $E_K = -72$, $E_A = -75$ (mV). All gating variables, $x \in \{n, m, h, a, b\}$, follow first-order kinetics:

$$\frac{dx}{dt} = \frac{x_\infty - x}{\tau_x}, \quad x_\infty = \frac{\alpha_x(V)}{\alpha_x(V) + \beta_x(V)}, \quad \tau_x = \frac{1}{\alpha_x(V) + \beta_x(V)}.$$

$x_\infty$ denotes the steady-state (in-)activation function of $x$, $\tau_x$ the time constant of (in-)activation, and $\alpha$ and $\beta$ opening and closing rates of $x$, respectively. Specifically,

$$\alpha_m = \frac{0.38 \cdot (V + 29.7)}{1 - \exp(-0.1 \cdot (V + 29.7))},$$

$$\beta_m = 15.2 \cdot \exp(-0.0556 \cdot (V + 54.7)),$$

$$\alpha_h = 0.266 \cdot \exp(-0.05 \cdot (V + 48)),$$

$$\beta_h = \frac{3.8}{1 + \exp(-0.1 \cdot (V + 18))},$$

$$\alpha_n = \frac{0.02 \cdot (V + 45.7)}{1 - \exp(-0.1 \cdot (V + 45.7))},$$

$$\beta_n = 0.25 \cdot \exp(-0.0125 \cdot (V + 55.7)),$$

$$\tau_a = 0.3632 + \frac{1.158}{1 + \exp(0.0497 \cdot (V + 55.96))},$$

$$a_\infty = \left[ \frac{0.0761 \cdot \exp(0.0314 \cdot (V + 94.22))}{1 + \exp(0.0346 \cdot (V + 1.17))} \right]^{1/3},$$

$$\tau_b = 1.24 + \frac{2.678}{1 + \exp(0.0624 \cdot (V + 50))},$$

$$b_\infty = \left( \frac{1}{1 + \exp(0.0688 \cdot (V + 53.3))} \right)^4.$$

All simulations were performed in Matlab (variable-order solver ode15 s [*Shampine and Reichelt, 1997*] with time step 0.01 ms).

## Temperature dependence of the model

The dependence on temperature $T(°C)$ was introduced to the model at the level of reversal potentials $E(T)$, peak conductances $\bar{g}_X(T)$, and time constants of (in-)activation $\tau_x(T)$. The Nernst equation defined the temperature dependence of reversal potentials:

$$E = \frac{RT}{zF}\ln\frac{[\text{ion outside}]}{[\text{ion inside}]} \Rightarrow E(T_0 + \Delta T) = E(T_0)\cdot\left(1+\frac{\Delta T}{T_0 + 273.15}\right).$$

$R$ denotes the universal gas constant, $z$ the valence of the considered ion, and $F$ the Faraday constant. $T_0 = 18°C$ sets the reference temperature, $\Delta T$ temperature differences. Temperature dependence of $\bar{g}_X(T)$ is determined by the choice of the parameter $Q_{10,\bar{g}_X}$:

$$\bar{g}_X \to \bar{g}_X(\Delta T) = \bar{g}_X\cdot Q_{10,\bar{g}_X}^{\frac{\Delta T}{10}}, \quad X \in \{Na, K, L, A\}.$$

For $\Delta T = 0$ each $\bar{g}_X(T)$ takes the value of its original definition at 18°C. For a given gating variable, temperature dependence of its opening and closing rates are identical and read:

$$\alpha_x \to \alpha_x(\Delta T) = \alpha_x\cdot Q_{10,x}^{\frac{\Delta T}{10}},$$

$$\beta_x \to \beta_x(\Delta T) = \beta_x\cdot Q_{10,x}^{\frac{\Delta T}{10}}, \quad x \in \{n, m, h, a, b\}.$$

Consequently,

$$\tau_x \to \tau_x(\Delta T) = \tau_x / Q_{10,x}.$$

Exploration of the parameter space

The model totals nine temperature-dependence parameters which span the parameter range explored: $Q_{10,\bar{g}_X}$, with $X \in \{L, Na, K, A\}$, and $Q_{10,x}$, with $x \in \{n, m, h, a, b\}$. Each parameter was sampled in four steps within realistic intervals (*Partridge and Connor, 1978*; *Hille, 2001*; *Tang et al., 2010*): $Q_{10,\bar{g}_X} \in [1.2, 2.0], Q_{10,x} \in [2.0, 4.0]$, resulting in a total number of $4^9$ models. Step currents of different amplitudes ($I_C \in [0.05, 0.6]$ $\mu A/mm^2$, varied in 0.05 $\mu A/mm^2$ steps) and 100 ms duration served as stimulus to the model neurons. Preceding and following a stimulus, periods of 50 ms were simulated without current stimulation. For each model, simulations were performed at 28°C and referenced with model behavior at 18°C to derive the temperature dependence.

## Quantification of f-I curve temperature dependence

Spike rates $f$ in response to $N = 12$ current amplitudes $I_C$ defined the f-I curve for a given temperature (spike detection threshold −30 mV). Temperature dependence of the f-I curve was quantified as the root-mean-squared difference between firing rates at the two temperatures across input currents, normalized by the average spike rate elicited at the lower temperature:

$$\text{RMSD} = \frac{\sqrt{\frac{1}{N}\sum_{i=1}^{N}\left(f_{i,T_{\text{cold}}}\left(I_{C,i}\right) - f_{i,T_{\text{hot}}}\left(I_{C,i}\right)\right)^2}}{\frac{1}{N}\sum_{i=1}^{N}f_{i,T_{\text{cold}}}\left(I_{C,i}\right)},$$

with $T_{\text{cold}} = 18°C$ and $T_{\text{hot}} = 28°C$. In agreement with the functional shape of type I spiking (*Ermentrout, 1996*), f-I curves for each $Q_{10}$ parameter combination were fit to a square root model,

$$r = \varphi\left(I_C\right) = A\cdot\sqrt{I_C - I_0},$$

where $A$ denotes slope and $I_0$ firing threshold of the f-I curve (quality of fit $R^2$ >0.97 for 99% of the models).

## Fisher information

For a spike generation process $f(I)$, Fisher information $J(I)$ is a measure of how accurately a particular input current $I$ can be decoded from the firing-rate response $f(I)$. It is formally defined as

$$J(I) = \int \left( \frac{\partial}{\partial I} \ln P(f \mid I) \right)^2 P(f \mid I) \, df,$$

with the conditional probability density of the spike rate given an input current, $P(f \mid I)$, characterizing the output noise (i.e., spike-rate variability). We consider two empirical response models for the spike rate density: Poissonian and input-independent Gaussian, reading

$$P_P(\tilde{f} \mid I) = \frac{(f(I) \cdot b)^{\tilde{f} \cdot b}}{(\tilde{f} b)!} \cdot \exp(-f(I) b) \text{ and}$$

$$P_G(\tilde{f} \mid I) = \frac{1}{\sqrt{2\pi\sigma^2}} \cdot \exp\left( -\frac{(\tilde{f} - f(I))^2}{2 \cdot \sigma^2} \right),$$

respectively. For the Poisson case, $b$ denotes the time bin during which a certain spike count $N_{sp}$ is observed. It is assumed to be sufficiently large so that $N_{sp} / b$ is well approximated by the mean firing rate $f(I)$. $\sigma^2$ denotes the variance of the Gaussian probability density. The corresponding Fisher information is given by

$$J_P(I) = \frac{(f'(I))^2}{f(I)} \text{ and}$$

$$J_G(I) = \frac{(f'(I))^2}{\sigma^2},$$

respectively. To compare Fisher information across different temperatures, it was averaged across a fixed interval of output firing rates $[f_{min}, f_{max}]$. Accordingly, the input current interval $[I_{min}, I_{max}]$ was computed for each model and temperature. Average Fisher information reads

$$\langle J \rangle = (I_{max} - I_{min})^{-1} \int_{I_{min}}^{I_{max}} dI \, J(I).$$

For low noise, the average Fisher information is a lower bound to the neuron's capacity to transmit information (**Kostal et al., 2013**),

$$C_{low} = \ln\left( \frac{\int dI \, J(I)}{\sqrt{2\pi e}} \right).$$

Exploiting the square-root shape of firing-rate curves, $f(I) = A \cdot \sqrt{I - I_0}$, and $f'(I) = A / 2 \cdot (I - I_0)^{-1/2}$, it follows that

$$\frac{1}{\Delta I} \equiv \frac{1}{I_{max} - I_{min}} = \frac{A^2}{f_{max}^2 - f_{min}^2}. \tag{1}$$

For the Poisson probability density Fisher information is given by

$$\langle J_P \rangle = \frac{1}{\Delta I} \int\limits_{I_{min}}^{I_{max}} dI \, A/4 \cdot (I - I_0)^{-3/2}$$

$$= \frac{1}{\Delta I} \cdot (-A/2) \left[ (I - I_0)^{-1/2} \right]_{I_{min}}^{I_{max}}$$

$$= \frac{1}{\Delta I} \cdot (-A^2/2) \left[ \left( A \sqrt{I - I_0} \right)^{-1} \right]_{I_{min}}^{I_{max}}.$$

With **Equation 1** it can be expressed as

$$\langle J_P \rangle = \frac{1}{\Delta I} \cdot (-A^2/2)(1/f_{max} - 1/f_{min})$$

$$= \frac{A^2}{f_{max}^2 - f_{min}^2} \cdot (-A^2/2)(1/f_{max} - 1/f_{min})$$

$$= A^4 \cdot \frac{1}{2 f_{max} f_{min} (f_{max} + f_{min})}.$$

For a Gaussian probability density we get

$$\langle J_G \rangle = \frac{1}{\Delta I} \int\limits_{I_{min}}^{I_{max}} dI \, A^2 / (2\sigma^2) \cdot \frac{1}{I - I_0}$$

$$= \frac{A^2 / (2\sigma^2)}{\Delta I} \cdot \left[ \ln(I - I_0) \right]_{I_{min}}^{I_{max}}$$

$$= \frac{A^2 / (2\sigma^2)}{\Delta I} \cdot \ln \left( \frac{I_{max} - I_0}{I_{min} - I_0} \right)$$

instead. Fisher information in this case reads

$$\langle J_G \rangle = \frac{A^4}{\sigma^2 (f_{max}^2 - f_{min}^2)} \cdot \ln \left( \frac{f_{max}}{f_{min}} \right).$$

Because only the slope of the firing-rate curve, $A$, is temperature-dependent in $\langle J_P \rangle$ and $\langle J_G \rangle$, the temperature dependence of Fisher information is given by

$$Q_{10}(\langle J_P \rangle) = Q_{10}(\langle J_G \rangle) = [Q_{10}(A)]^4.$$

For the average value across a fixed output interval [$f_{min}$, $f_{max}$] Fisher information is invariant to shifts of the threshold. A heating-induced increase in the accuracy of a decoder hence requires an increase in slope of the firing-rate curve, that is $Q_{10}(A) > 1$. This is true for all spike-generation models considered.

## Measures of energy consumption during spiking and rest

### Spiking cost

Energy consumption per spike was quantified as the total sodium current (also termed sodium load) (**Hasenstaub et al., 2010**; **Sengupta et al., 2010**) between stimulus onset $t_{start}$ and 20 ms post stimulus offset $t_{stop}$, divided by the number of spikes elicited during this period, $N_s$:

$$Na^+ load / spike = \int\limits_{t_{start}}^{t_{stop} + 20ms} I_{Na} dt / N_s,$$

Considering that the *Na-K*-ATPase consumes one ATP molecule per 3 $Na^+$ ions, this quantity is proportional to the number of ATP molecules per spike. The $Q_{10}(Na^+$ load/spike) averaged across all input currents was used for further analysis (and referred to as $Q_{10}$(spiking cost)).

In addition, a measure of energy efficiency based on the separation of charges, that is the fraction of the sodium current that was not counterbalanced by a simultaneous potassium current (*Crotty et al., 2006*), was implemented. As for estimation of the sodium-current-based cost, the stimulus period and the following 20 ms were evaluated. Note that the potassium current in the model comprised two components, $I_{K,total} = I_K + I_A$. Temperature effects on energy efficiency (estimated by the corresponding $Q_{10}$ values) were highly similar to those on (the inverse of) spiking energy consumption. Finally, both measures—the current-based cost and the charge-separation-based energy efficiency— were also implemented based on the potassium current instead of the sodium current. In other words, spiking cost was additionally quantified by the total potassium current, energy efficiency based on the fraction of the potassium current that was not counterbalanced by the sodium current.

## Resting cost
The resting potential $V_r$ of the Connor–Stevens model is given by

$$V_r = \frac{E_A \cdot g_A(V_r) + E_L \cdot \overline{g}_L + E_K \cdot g_K(V_r) + E_{Na} \cdot g_{Na}(V_r)}{\overline{g}_L + g_A(V_r) + g_K(V_r) + g_{Na}(V_r)}. \quad (2)$$

The numerical solution of this expression for $V_r$ was used to evaluate the sodium current at rest,

$$I_{Na,rest} = g_{Na}(V_r) \cdot (E_{Na} - V_r),$$

which defines the sodium-current-based cost of the resting potential (assuming the current is proportional to the activity of the *Na-K*-ATPase at rest). Likewise, the potassium-current-based cost is defined by

$$I_{K,rest} = g_K(V_r) \cdot (E_K - V_r) + g_A(V_r) \cdot (E_A - V_r).$$

As the resting state is steady, the cost quantities do not depend on the gating variables' (in-) activation time constants and, consequently, the corresponding $Q_{10}$ values do not have an impact on $Q_{10}$(resting cost).

## Sensitivity analysis
Sensitivity analysis was performed in the parameter space spanned by the nine temperature-dependence parameters (each dimension sampled by four values). To quantify global impact of one parameter on a given observable (like RMSD), we evaluated the distribution of point-wise changes in the observable along the dimension of a specific parameter. In total, for each parameter, $3 \cdot 4^8$ changes between neighboring points along the corresponding dimension need to be considered. These define a distribution of changes, whose median is indicative of the global *impact* of this parameter on the observable. The distribution's 25% and 75% percentiles are indicated as error bars (see, e.g., *Figure 2C*). For each observable, impact values were normalized to give unity when summed across all nine parameters. The sign of the impact provides an estimate of the qualitative influence of the parameter on the observable, that is whether an increase in the parameter value leads to an increase or decrease in the observable. We considered an impact reliable if both percentiles (25% and 75%) had the same sign as the impact.

Note that our impact evaluation constitutes a *global* sensitivity analysis, comparable to the Morris one-at-a-time method (*Morris, 1991*). In contrast to the latter approach, we use a full factorial (grid) set of inputs instead of a random one. Moreover, our measure is based on the median instead of the mean of a distribution of differences in the observable. Yet, the interpretation of a high absolute impact is comparable to that of a high (absolute) mean elementary effect (the sensitivity measure in *Morris 1991*), as is the interpretation of a large interquartile range of the difference distributions to a large standard deviation of the elementary effect.

## RMSD minimization with a genetic algorithm
Alternatively to the coarse parameter grid search, we also used optimization by a genetic algorithm (*Mitchell, 1998*) to validate the minimum RMSD. To this end, the turboGA function was used (Matlab file exchange; settings: population size 1000, 150 generations, 8 bit discretization, initial conditions uniformly random). For the $Q_{10}$ parameter range used in the main article, the minimum RMSD identified by the genetic algorithm was very close to the minimum value on the grid (0.21 for the genetic algorithm, 0.22 on the grid). On average the coordinates of the grid-based minimum deviated 5% from the coordinates of the genetic algorithm-based minimum.

## Dimensional stacking

Dimensional stacking is a method to visualize high-dimensional data, that is an observable $f$ as a function of $N$ parameters, $p_1, \ldots, p_N$, evaluated at a discrete set of parameter values. The method is described in detail, for categoric observables, in *LeBlanc et al. (1990)* and *Taylor et al. (2006)*. Mainly, the method maps the $N$-dimensional data to a two-dimensional representation by iteratively slicing the data in one dimension and stacking the slices in 2D (*Figure 2L*). In this representation, the position of each pixel in the two-dimensional image corresponds to one parameter combination, and its color encodes the value of the observable. The image has $N$ axes of different scales, each associated with one parameter. Visual informativeness of a dimensional stacking image crucially depends on the order in which the dimensions are stacked, that is, the axes order. The parameter dimensions associated with larger variability in the observable should be assigned larger-scale axes; those with lowest variability the small-scale axes. Sorting the axes with respect to their impact on the observable prior to dimensional stacking hence leads to a visually informative image, where color changes can be easily related to changes of the observable with individual parameters. For this study, we used the ranking of absolute impact scores (described above) to define the 'optimal' stack order, extending the optimization method described in *LeBlanc et al. (1990)* and *Taylor et al. (2006)*.

## Model robustness

The sensitivity analysis was performed with the Connor–Stevens model with original parameters for peak conductances $\bar{g}_{Na}$, $\bar{g}_K$, $\bar{g}_A$, and $\bar{g}_L$ at the colder temperature (*Dayan and Abbott, 2005*). To test that our results are robust and do not strongly depend on this specific parameter choice, we additionally performed the whole sensitivity analysis for 24 models with peak conductances of the reference model perturbed by ±20% (8 models with one individual peak conductance lowered or raised by 20%; 16 models with all combinations of the four conductances either lowered or raised by 20%). The impacts for those models are summarized in *Figure 2E* and *Figure 3F,G* (individual changes in peak conductances represented by black symbols, combined changes by gray ones). Note that for computational efficiency only three values per parameter (instead of four) were taken (for perturbed models as well as the reference model, as presented in *Figure 2E*). The fraction of models with RMSD < 0.5 (across the temperature dependence parameter space) was 18% in the original Connor–Stevens model. Variations in peak conductances did not change this finding much: for each perturbed model 15–19% of its temperature dependence combinations gave RMSD < 0.5.

For completeness, we also checked that a structurally different vertebrate model with type I dynamics (Traub-Miles, *Traub et al., 1991*), as defined in *Benda 2002*) was able to display temperature compensation despite a substantial temperature dependence of individual conductances (same range of temperature parameters as in the Connor–Stevens models, *Figure 2—figure supplement 1*). The search for the lowest temperature dependence within the parameter space (sodium and potassium kinetics $Q_{10}(m)$, $Q_{10}(h)$, $Q_{10}(n)$, and the peak conductances of sodium, potassium and leak $Q_{10}(\bar{g}_{Na})$, $Q_{10}(\bar{g}_K)$, $Q_{10}(\bar{g}_L)$) was performed based on the genetic algorithm described above. As the model operates at 32°, we checked both heating and cooling the model by 10°.

## Reverse-engineering the transduction and its temperature dependence

For a given pair of a receptor neuron response $r = \rho(I_{dB})$ and a spike generation model $r = \varphi(I_C)$, the transduction function (current $I_C$ vs sound intensity $I_{dB}$) can be inferred. We assumed a sigmoidal shape

of the transduction function, $I_C = \theta(I_{dB}) = I_{C,sat} \Big/ \left( 1 + \exp\left( -\dfrac{I_{dB} - I_{50,\theta}}{w_\theta} \right) \right)$, with transduction saturation current $I_{C,sat}$, half-maximum sound intensity $I_{50,\theta}$, and dynamic-range width $w_\theta$. Further, we chose representative parameters for the receptor neuron response. To this end, the median cold temperature, $\bar{T}_c$, the median receptor neuron response parameters at cold temperature, $\bar{p}_\rho \in \{\bar{r}_{sat}, \bar{I}_{50,\rho}, \bar{w}_\rho\}$, and the median temperature dependencies of the three receptor neuron response parameters, $\bar{Q}_{10}(p_\rho)$, were determined from the experimental data. Using these, receptor neuron response parameters were inferred for temperatures of 18 and 28° (the temperatures at which spike generation simulations were per-

formed), according to $p_{\rho,T} = \bar{p}_\rho \cdot (\bar{Q}_{10}(p_\rho))^{\frac{T - \bar{T}_c}{10}}$. The resulting representative receptor neuron responses were used as objective functions for reverse engineering of the transduction curve. To infer the three parameters characterizing the optimal transduction curve for a given spike generation model $r = \varphi(I_C)$, we computed

$$r = \varphi(\theta(I_{dB})) = A \cdot \sqrt{I_{C,\text{sat}} \Big/ \left(1 + \exp\left(-\frac{I_{dB} - I_{50,\theta}}{w_\theta}\right)\right)} - I_0 \quad .$$

The transduction parameters were chosen such that they minimized the root mean squared error between $\varphi\left(\theta\left(I_{dB}\right)\right)$ and the representative receptor neuron response $\rho(I_{dB})$:

$$\theta\left(I_{dB}, \text{ optimal}\right) = \arg\ \min_{\theta(I_{dB})}\left\{\int |\rho\left(I_{dB}\right) - \varphi\left(\theta\left(I_{dB}\right)\right)|^2\ dI_{dB}\right\}$$

This fitting procedure was repeated for all hot and cold spike generation processes, and the temperature coefficients for the three transduction parameters were computed. The transduction parameters at the reference temperature (18°C) were: $I_{C,\text{cold}}\left(I_{dB}\right) = 0.40\,\mu\text{A}/\text{mm}^2$.

## Acknowledgements

This work was funded by grants from the Federal Ministry of Education and Research, Germany (01GQ1001A, 01GQ0901, 01GQ0972) and the Deutsche Forschungsgemeinschaft (SFB618, GRK1589/1).

## Additional information

### Funding

| Funder | Grant reference number | Author |
| --- | --- | --- |
| Federal Ministry of Education and Research (BMBF) | 01GQ0901 | Frederic A Roemschied, Susanne Schreiber |
| Deutsche Forschungsgemeinschaft (DFG) | SFB 618 | Monika JB Eberhard, Bernhard Ronacher, Susanne Schreiber |
| Federal Ministry of Education and Research (BMBF) | 01GQ0972 | Jan-Hendrik Schleimer, Susanne Schreiber |
| Federal Ministry of Education and Research (BMBF) | 01GQ1001A | Bernhard Ronacher, Susanne Schreiber |
| Deutsche Forschungsgemeinschaft (DFG) | GRK 1589/1 | Frederic A Roemschied, Bernhard Ronacher, Susanne Schreiber |

The funders had no role in study design, data collection and interpretation, or the decision to submit the work for publication.

### Author contributions

FAR, Implemented the mathematical modeling, Wrote the article, Conception and design, Analysis and interpretation of data; MJBE, Acquisition of data, Analysis and interpretation of data, Drafting or revising the article; J-HS, BR, Conception and design, Analysis and interpretation of data, Drafting or revising the article; SS, Wrote the article, Conception and design, Analysis and interpretation of data

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
