## [Decision Letter]

Thank you for sending your work entitled “Cell-intrinsic mechanisms of temperature compensation in a sensory receptor neuron” for consideration at *eLife*. Your article has been favorably evaluated by a Senior editor and 3 reviewers, one of whom, Ronald L Calabrese, is a member of our Board of Reviewing Editors.

The Reviewing editor and the other reviewers discussed their comments before we reached this decision, and the Reviewing editor has assembled the following comments to help you prepare a revised submission.

The authors describe experiments showing temperature insensitivity in locust auditory receptor neurons (Q10< 1.5) and then pursue a modeling approach to explain these results. They concentrate on the spiking mechanisms itself and use an ensemble modeling approach to identify model neurons that will show comparable temperature insensitivity while expressing membrane currents with normal Q10s. These modeling experiments identify key processes (parameters) that give rise to the temperature 'compensation' and show that well-compensated models show a shift in threshold that partially cancels the shift in F-I slope that all models show. They also address the issues of information transfer and metabolic cost as temperature increases and find no restrictions for compensated models. They then use a reverse engineering method to address temperature compensation in the transduction mechanisms (receptor potential). The finding are of general interest because many neurons, including peripheral mammalian neurons, experience temperature changes and these studies show that such changes are easily accommodated/compensated by systems of membrane currents with normal Q10s.

The three reviewers are from different backgrounds and have different overlapping expertise; their reviews reflect their different backgrounds and expertise, but they all agree that four major revisions are needed.

1) While conciseness is a virtue it should not sacrifice clarity and completeness. We encourage the authors to expand their style in Introduction, Results, and even Methods to add more rationale and fuller explanations of what they did and how the models were constructed and used. In particular the section “Model-based inference of the auditory transduction function” was very difficult to follow. We suggest that the authors expand this section and fully explain what they did, how they did it, and why they did it.

2) Like prior studies by Laughlin and others on the metabolic cost of neuronal signaling, the authors use Na flux as a proxy for ion pump activity and as their measure of metabolic cost during spiking and at rest. The authors should provide clear rationale for why the cost of other ions were excluded. This seems particularly relevant in light of the authors' finding that the properties of Na channels, more than those of other channels, influence energy efficiency in these model neurons. The model used only includes Na and K currents and so other ions cannot be considered and the cost of K is tied to the cost of Na by the Na/K pump but this and the inherent limitations it imposes on the conclusion should be made explicit.

3) One reviewer crystallized concerns about the experimental component of the work: “My concerns are with the acquisition of the biological data, particularly with respect to temperature control and measurement…” The authors should make their experimental procedures crystal clear (in both Results and in Methods) and should address each of the questions posed by this reviewer. They should also make arguments about how much error there would have to be to affect their conclusion.

4) The authors should apply appropriate statistical tests to the biological data in all cases where temperature appears to affect a characteristic, e.g., spike width, so that the reader is confident that the temperature change is large enough to measure valid Q10s.

Other issues to address:

*Reviewer*
*#1:*

The section “Model-based inference of the auditory transduction function” was very difficult to follow. This reviewer is a computational neuroscientist but I am unfamiliar with reverse engineering techniques, and I was not given enough help in reasoning through the experiments and data as would be necessary for the general reader in neuroscience. I suggest that the authors expand this section and fully explain what they did, how they did it, and why they did it. The Methods section helps a bit but only very marginally and is very technically written. Think of the general reader here.

*Reviewer*
*#2:*

Like prior studies by Laughlin and others on the metabolic cost of neuronal signaling, the authors use Na flux as a proxy for ion pump activity and as their measure of metabolic cost during spiking and at rest. I have always been puzzled by this choice – why exclude the cost of other ions? I assume there are good reasons for this choice that I am simply not aware of, so I suggest that the authors include a comment on this question. This seems particularly relevant in light of the authors' finding that the properties of Na channels, more than those of other channels, influence energy efficiency in these model neurons. This result makes me wonder if the outcome would have been different (with other ion channels playing a bigger role in energy efficiency) if metabolic cost had been defined in a less Na-centric way.

Another comment concerns the tests for robustness of their findings to the particular choice of model neuron that the authors describe in the paper. In the Results section we learn that the authors tested for robustness of their results by checking whether a relatively modest (20%) variation of parameters of the Connors-Stevens model used here changes the overall conclusions, and are assured that it does not. While this test for robustness is laudable, I was much more convinced that the findings are robust by a very short comment, hidden in the Methods section, in which the authors state that they also found temperature compensation of firing properties despite temperature dependence of ion channels in a structurally different Traub-Miles model neuron. I recommend that the authors consider giving this latter result a more prominent place in the paper, because it could potentially (since I don't know the details) make the case for generality of the results much stronger.

*Reviewer*
*#3:*

The work is novel, interesting and well presented. Though the linkage between the biological neurons and the models is quite tenuous, I have no problems with the computational aspects of the paper. My concerns are with the acquisition of the biological data, particularly with respect to temperature control and measurement. This is important because if the temperature change at the receptor ending is over-estimated then the Q10s will be under-estimated and there may not be an interesting biological phenomenon (the surprisingly low Q10) in need of explanation. The preparation was heated by placing it on a Peltier element i.e. a localized heat source underneath the locust. Heat conduction from that source will vary through the various tissues, cuticle and saline and I would expect that the preparation was not at a uniform temperature throughout. The GTF 300 thermal probe has, according to the manufacturer's website, a diameter of 1mm (assuming that it hadn't been modified) which is relatively large with respect to distances inside the thoracic/abdominal cavity. I wonder if the authors could provide more information in the Methods that might increase confidence that the temperature range reported and used to calculate Q10 was the range that the receptor ending actually experienced. For example, how was the temperature at the attachment site of the receptor neurons measured using the GTF 300 probe? How variable was the temperature at different sites in the preparation (tympanum, saline, ganglion)? What did the calibration curve look like and was a calibration curve generated for every preparation that provided data or from other preparations; how consistent was it between preparations?

Action potential width was measured at the recording site in the metathoracic ganglion but it is being associated with firing frequency generated at the tympanum. Did these two sites experience the same temperature changes? Given that the density and diversity of conductances can be quite different at different locations of a neuron, do we know whether action potential parameters at these two sites are the same?

The sample size of 9 neurons in 8 preparations is relatively small. I assume that each preparation provided at least one neuronal recording and one provided two, but this is not explicitly stated. According to Figure 1, one or more of the neurons had a Q10 for spike rate at half max greater than 2.5 suggesting that these measures were quite variable. Did the authors perform any statistical comparisons to test significance?

It is unsurprising to me that metabolic cost was defined as Na load and then the computational analysis revealed that parameters of the Na conductance had the greatest effect on Q10 of metabolic cost. The authors might comment on this.

---

## [Author Response]

*1) While conciseness is a virtue it should not sacrifice clarity and completeness. We encourage the authors to expand their style in Introduction, Results, and even Methods to add more rationale and fuller explanations of what they did and how the models were constructed and used. In particular the section “Model-based inference of the auditory transduction function” was very difficult to follow. We suggest that the authors expand this section and fully explain what they did, how they did it, and why they did it*.

We have expanded all paragraphs, with a main focus on Results and Methods. In particular the section “Model-based inference of the auditory transduction function” has been rewritten. We have also added a total of six supplemental figures (using the figures supplements in the *eLife* format) for more detailed information on the analyses.

*2) Like prior studies by Laughlin and others on the metabolic cost of neuronal signaling, the authors use Na flux as a proxy for ion pump activity and as their measure of metabolic cost during spiking and at rest. The authors should provide clear rationale for why the cost of other ions were excluded. This seems particularly relevant in light of the authors' finding that the properties of Na channels, more than those of other channels, influence energy efficiency in these model neurons. The model used only includes Na and K currents and so other ions cannot be considered and the cost of K is tied to the cost of Na by the Na/K pump but this and the inherent limitations it imposes on the conclusion should be made explicit*.

We extended the analysis and included additional measures of energy efficiency/cost). The results for measures of charge separation (for spiking cost) are now shown. Moreover, costs are now also estimated based on the potassium currents (results presented in supplemental figures to Figure 3). Despite minor quantitative changes, the main findings hold for all measures. As pointed out by the referees, costs arising from the flow of other ions can contribute to the metabolic cost in living cells, but are not included in the model. We assume that the major cost in our study arises from the activity of the Na-K-ATPase.

*3) One reviewer crystallized concerns about the experimental component of the work: “My concerns are with the acquisition of the biological data, particularly with respect to temperature control and measurement…” The authors should make their experimental procedures crystal clear (in both Results and in Methods) and should address each of the questions posed by this reviewer. They should also make arguments about how much error there would have to be to affect their conclusion*.

Below we are addressing all concerns of the reviewer and provide more detail on the experimental procedures. The calibration curve is now included as a supplement to Figure 1. The errors in temperature measurement would indeed have to be large in order to substantially affect our results. To moderately change a firing-rate Q10 value from 1.5 to 1.6, the temperature error would have to be on the order of 1 degree Celsius or more. Our error ranges, as can be seen from the calibration curve, are much lower.

*4) The authors should apply appropriate statistical tests to the biological data in all cases where temperature appears to affect a characteristic, e.g., spike width, so that the reader is confident that the temperature change is large enough to measure valid Q10s*.

As suggested, we have performed additional statistical tests. Tests of differences between warm and cold spike rates are now provided as a function of sound intensity (a corresponding table can be found in a supplement to Figure 1). Also, a rank sum test was used to test whether the Q10 values of the individual characteristics (spike saturation rate, etc) are significantly different from 1.0, 1.5, and 2.0. These data are also provided.

*Other issues*
*to address:*

Reviewer #1:

*The section “Model-based inference of the auditory transduction function” was very difficult to follow. This reviewer is a computational neuroscientist but I am unfamiliar with reverse engineering techniques, and I was not given enough help in reasoning through the experiments and data as would be necessary for the general reader in neuroscience. I suggest that the authors expand this section and fully explain what they did, how they did it, and why they did it. The Methods section helps a bit but only very marginally and is very technically written. Think of the general reader here*.

The corresponding paragraph was completely rewritten and we hope that it is now better understandable. Where appropriate, also Introduction and Results have been extended. The Methods section now provides more details on additional techniques used (like the charge separation measure and the genetic algorithm).

Reviewer #2:

*Like prior studies by Laughlin and others on the metabolic cost of neuronal signaling, the authors use Na flux as a proxy for ion pump activity and as their measure of metabolic cost during spiking and at rest. I have always been puzzled by this choice – why exclude the cost of other ions? I assume there are good reasons for this choice that I am simply not aware of, so I suggest that the authors include a comment on this question. This seems particularly relevant in light of the authors' finding that the properties of Na channels, more than those of other channels, influence energy efficiency in these model neurons. This result makes me wonder if the outcome would have been different (with other ion channels playing a bigger role in energy efficiency) if metabolic cost had been defined in a less Na-centric way*.

We address this criticism and have now also performed the metabolic cost analysis based on potassium current flow. In addition, we use measures of charge separation (for spiking cost), which quantify the degree of overlap between sodium and potassium currents and are hence conceptually different from a cost estimated by pure current flow of one ion type. Nevertheless, also charge separation is biased towards one ion type, as it quantifies the fraction of the Na current that is not counterbalanced by a simultaneous potassium current. This is why we now characterize the “alternative scenario” as well, i.e. the amount of total potassium current that is not counterbalanced by a simultaneous Na current. Despite quantitative differences between the individual measures, temperature dependence of Na channel inactivation remains the main player in all cases and exerts the largest influence on metabolic cost (or energy efficiency). Besides, the main player for firing rate compensation (i.e., the temperature dependence of potassium channel kinetics, Q10(n)) has only little impact on the potassium-current-based metabolic cost.

*Another comment concerns the tests for robustness of their findings to the particular choice of model neuron that the authors describe in the paper. In the Results section we learn that the authors tested for robustness of their results by checking whether a relatively modest (20%) variation of parameters of the Connors-Stevens model used here changes the overall conclusions, and are assured that it does not. While this test for robustness is laudable, I was much more convinced that the findings are robust by a very short comment, hidden in the Methods section, in which the authors state that they also found temperature compensation of firing properties despite temperature dependence of ion channels in a structurally different Traub-Miles model neuron. I recommend that the authors consider giving this latter result a more prominent place in the paper, because it could potentially (since I don't know the details) make the case for generality of the results much stronger*.

We have taken up this suggestion and now present the results for the Traub- Miles model in a supplemental figure (to Figure 2). For this model we only performed the (computationally more efficient) search for a minimal RMSD with a genetic algorithm, identifying the most temperature invariant temperature parameter set. The results resolve that also for this model a temperature compensation with RMSD values on the order of 0.35-0.60 can be achieved.

Reviewer #3:

*The work is novel, interesting and well presented. Though the linkage between the biological neurons and the models is quite tenuous, I have no problems with the computational aspects of the paper. My concerns are with the acquisition of the biological data, particularly with respect to temperature control and measurement. This is important because if the temperature change at the receptor ending is over-estimated then the Q10s will be under-estimated and there may not be an interesting biological phenomenon (the surprisingly low Q10) in need of explanation*.

We have taken care to avoid an overestimation of the temperature change. Tissue temperature was derived from the calibration curve in such a way, that we explicitly favor an underestimation of the temperature change and hence bias our results towards higher Q10 values. The calibration curve is now included. In particular, temperature changes were estimated based on the temperature at the onset of a recording. During a 40-seconds-long recording, we cannot exclude that temperature further decreased and hence the effective temperature change was larger than assumed by our conservative approach. This way, however, an underestimation of Q10 values is excluded. For more details on the experimental procedures see responses below.

*The preparation was heated by placing it on a Peltier element i.e. a localized heat source underneath the locust. Heat conduction from that source will vary through the various tissues, cuticle and saline and I would expect that the preparation was not at a uniform temperature throughout. The GTF 300 thermal probe has, according to the manufacturer's website, a diameter of 1mm (assuming that it hadn't been modified) which is relatively large with respect to distances inside the thoracic/abdominal cavity. I wonder if the authors could provide more information in the Methods that might increase confidence that the temperature range reported and used to calculate Q10 was the range that the receptor ending actually experienced. For example, how was*
*the temperature at the attachment site of the receptor neurons measured using the GTF 300 probe?*

The diameter of the insulated wire of the GTF 300 probe is 1mm but the actual measuring tip consists of two twisted wires (without insulation) with a diameter of 0.2mm each, with the welded tip of the probe having a total diameter of 0.30mm (see https://greisinger.de/files/upload/en/produkte/kat/k13_127_EN_oP.pdf).

The measuring tip of the GTF 300 probe was placed in the saline-filled thoracic cavity, as close as possible to the tympanal membrane (less than 1mm distance between measuring tip and tympanum).

*How variable*
*was the temperature at different sites in the preparation (tympanum, saline, ganglion)?*

Temperature at the measuring site next to the tympanum was not significantly different from the temperature at the ganglion (ca. 6-9mm distance between tympanum and ganglion, depending on the size of the locust; Wilcoxon rank sum test: Z= - 0.10, p=0.92); the whole preparation was well filled with ringer’s solution which helped to establish a constant temperature throughout the whole preparation. Within the time window used for the second recording (3-5.5 min after start of cooling), the mean temperature difference between tympanum and ganglion was 0.19°C. Please note that in order to change the Q10 of firing rate by 0.1 (e.g. from 1.5 to 1.6) a temperature deviation of more than 1 degree Celsius would be required for the mean spike rates recorded in the study.

*What did the calibration curve look like and was a calibration curve generated for every preparation that provided data or from other preparations; how consistent was*
*it between preparations?*

The calibration curve (supplement to Figure 1) was generated using four separate preparations, repeating the cooling steps and measurements three times each. The curve was consistent between preparations; different preparations than those used for recording were taken to establish the calibration curve, because a direct measurement was not possible: First, parallel temperature measurements during electrophysiological recordings were not possible due to substantially increased noise and a high risk of tissue damage. Second, temperature measurements after recording would only have been possible in the dissected and hence strongly modified system (after preservation for staining-based cell identification).

*Action potential width was measured at the recording site in the metathoracic ganglion but it is being associated with*
*firing frequency generated at the tympanum. Did these two sites experience the same temperature changes?*

Yes. Within the time window used for the second recording (3-5.5 min after start of cooling), the mean temperature difference between tympanum and ganglion was 0.19°C.

*Given that the density and diversity of conductances can be quite different at different locations of a neuron, do we know whether action potential parameters at these two*
*sites are the same?*

Action potentials were measured at the same location at both temperatures. We assume that spike initiation happens at the same position at both temperatures and spikes are hence generated by the same set of conductances. Firing rates could in principle be affected by other conductances on the way from the initation site to the ganglion. However, as distances travelled are short (6-9mm), we would not expect a large effect in firing rate, even if conductances of different kinetics were present. In addition, those hypothetical conductances would be subject to the same temperature changes.

*The sample size of 9 neurons in 8 preparations is relatively small. I assume that each preparation provided at least one neuronal recording and one provided two, but this is not explicitly stated*.

Yes, one of the preparations provided recordings from two neurons.

*According to*
Figure 1*, one or more of the neurons had a Q10 for spike rate at half max greater than 2.5 suggesting that these measures were quite variable*
*. Did the authors perform any statistical comparisons to test significance?*

We now provide statistical tests. First, we test the original firing-rate data. We ask whether mean firing rates at the two temperatures are different, separately analyzed for each sound intensity level (supplement to Figure 1). These data show that differences only are significant at larger sound intensities. For lower intensities noise is too large to allow for a clear differentiation. This also indicates that the temperature dependence of the saturation-firing rate is more reliable than the rate estimated at half-maximum sound intensity. Second, we tested whether the Q10 values (presented in Figure 1) are significantly different from 1.0, 1.5, and 2.0. Again, Q10 values of the half-maximum sound intensity rates are statistically weaker than the saturation-rate based Q10 values. The latter are significantly different from 1.0 as well as 2.0. For the half-maximum intensity values statistical significance is (strictly) only given for difference from 1.0, but not 2.0. These effects arise from two outliers, though, which is now visible in the graphical representation of the data in Figure 1.

*It is unsurprising to me that metabolic cost was defined as Na load and then the computational analysis revealed that parameters of the Na conductance had the greatest effect on Q10 of metabolic cost. The authors might comment on this*.

We kindly refer to our reply to reviewer 2 above, who made a similar comment. In short, we implemented three additional measures (potassium-current based cost and two measures of charge separation). Despite quantitative differences between the individual measures, temperature dependence of Na channel inactivation remained the main player in all cases and exerted the largest influence on metabolic cost (or energy efficiency). Besides, the main player for firing rate compensation (i.e., the temperature dependence of potassium channel kinetics, Q10(n)) had only little impact on the potassium-current-based metabolic cost.